# Radioprotective Effect of Flavonoids on Ionizing Radiation-Induced Brain Damage

**DOI:** 10.3390/molecules25235719

**Published:** 2020-12-03

**Authors:** Qinqi Wang, Chenghao Xie, Shijun Xi, Feng Qian, Xiaochun Peng, Jiangrong Huang, Fengru Tang

**Affiliations:** 1Laboratory of Oncology, Center for Molecular Medicine, School of Basic Medicine, Health Science Center, Yangtze University, Jingzhou 434023, China; 13545641579@163.com (Q.W.); akstella@163.com (C.X.); x455066483@163.com (S.X.); 2Department of Pathophysiology, School of Basic Medicine, Health Science Center, Yangtze University, Jingzhou 434023, China; 3Department of Physiology, School of Basic Medicine, Health Science Center, Yangtze University, Jingzhou 434023, China; qianfeng19781@hotmail.com; 4Department of Integrative Medicine, School of Health Sciences, Health Science Center, Yangtze University, Jingzhou 434023, China; 5Radiation Physiology Laboratory, Singapore Nuclear Research and Safety Initiative, National University of Singapore, 1 CREATE Way #04-01, CREATE Tower, Singapore 138602, Singapore

**Keywords:** flavonoids, radiation, oxidative stress, ROS, DNA damage, inflammation

## Abstract

Patients receiving brain radiotherapy may suffer acute or chronic side effects. Ionizing radiation induces the production of intracellular reactive oxygen species and pro-inflammatory cytokines in the central nervous system, leading to brain damage. Complementary Chinese herbal medicine therapy may reduce radiotherapy-induced side effects. Flavonoids are a class of natural products which can be extracted from Chinese herbal medicine and have been shown to have neuroprotective and radioprotective properties. Flavonoids are effective antioxidants and can also inhibit regulatory enzymes or transcription factors important for controlling inflammatory mediators, affect oxidative stress through interaction with DNA and enhance genomic stability. In this paper, radiation-induced brain damage and the relevant molecular mechanism were summarized. The radio-neuro-protective effect of flavonoids, i.e., antioxidant, anti-inflammatory and maintaining genomic stability, were then reviewed. We concluded that flavonoids treatment may be a promising complementary therapy to prevent radiotherapy-induced brain pathophysiological changes and cognitive impairment.

## 1. Introduction

Natural and man-made ionizing radiation is widespread in our environment. It has been well-documented that high-dose radiation exposure, such as brain radiotherapy, causes brain damage and cognitive impairment [1,2]. Sustained exposure to low doses of ionizing (e.g., repeated X-rays and computed tomography (CT) scans) and non-ionizing (e.g., mobile devices) radiation may also induce significant brain neuropathological changes and subsequent neurological and neuropsychological disorders [3]. Almost all human tissues are sensitive to the carcinogenic effects of radiation [4]. Even electromagnetic field exposure may induce damage to the nervous system [5]. Prenatal exposure to radiation increases the vulnerability of developing fetuses with brain damage, leading to microcephaly, mental retardation, epilepsy, brain tumors and aging [1,6]. Postpartum exposure can also have harmful effects on the nervous system, significantly reducing neurogenesis, glial formation and endothelial growth, leading to cognitive impairment and neuroinflammation in aging [2,7].

With an increasing incidence of head and neck malignancies, radiotherapy plays a key role in the treatment of head and neck cancers, either alone or in combination with surgery, chemotherapy and molecular targeting agents. However, it can cause damage to normal brain tissue, leading to serious health consequences, such as neuroinflammation, neuronal loss, impairment of neurogenesis and subsequent cognitive impairment [1,2,3,7,8]. In spite of the tremendously improved dose conformity of intensity-modulated radiotherapy (IMRT) and fast ion-beam radiation, there is still a need to use radioprotective drugs to protect at-risk organs. This is because high-energy radiation is necessary for ion-beam radiotherapy, and the possible risk of high linear energy transfer (LET) radiation in the surrounding normal tissue may be of more general concern, even though the absolute dose level is reduced [9,10]. Complementary treatment with natural products with less side effects has become necessary as many natural flavonoids, including apigenin, (−)-epigallocatechin gallate (EGCG), genistein and quercetin, may not only increase the radiosensitivity of cancer cells but also protect normal tissues from radiation-induced damage [11,12].

Flavonoids are plant-derived compounds with antioxidant, anti-inflammatory and radio-neuro-protective effects [13]. Flavonoids and their metabolites can cross the blood–brain barrier (BBB) [14], which is made up of capillary endothelial cells and basement membrane, neuroglial membrane and glial podocytes, i.e., projections of astrocytes [15], and reach brain cells to reduce brain damage and alleviate neurodegenerative diseases and cognitive impairment [16,17,18,19]. The neuroprotective mechanisms of flavonoids may include anti-oxidation, anti-apoptosis, anti-neuroinflammation and regulation of various intracellular and extracellular targets. Recent in vitro experiments have shown that EGCG and its metabolites, such as theaflavins, can enter the brain parenchyma through the BBB, induce neuritogenesis and have a potential neuroprotective effect [20]. Troxerutin, a natural flavonoid with a high capability to pass through the BBB, can prevent rotenone-induced retinal neurodegeneration [21]. Quercetin, a flavonoid, has obvious neuroprotective effects on radioactive brain injury [22]. Radiotherapy of patients with brain tumors leads to learning and memory impairments. Although the pathological cascade of cognitive deficits remains unknown, reduced hippocampal neurogenesis may be involved in the pathogenesis. It has been observed in cell and animal experiments that baicalin can reduce radiation-induced hippocampal neuronal damage by regulating oxidative stress and upregulating brain-derived neurotrophic factor (BDNF)-phosphor-(cyclic-AMP response element-binding protein) (p-CREB) signal transduction. In addition, baicalein (5,6,7 trihydroxyflavone) can prevent spatial learning and memory impairment caused by radiotherapy [23]. In view of the widespread existence of radiation and its tremendous damage to the human body, it is necessary to find a safe and effective radiation protection agent, and flavonoids are suitable candidates.

## 2. Mechanisms of Radiation-Induced Brain Injury

The brain is the main dose-limiting organ for patients receiving radiation therapy under various conditions. Radiation-induced brain injury is a continuous dynamic process which can be generally divided into three stages: (1) acute reactions, which occur within 2 weeks after the onset of radiation, including nausea and vomiting, headache, fatigue and increased neurological symptoms and signs; (2) early delayed reactions, which occur from 2 weeks to 6 months after irradiation and may be related to a transient demyelinating processes associated with the breakdown of the BBB or with selective oligodendroglia cell dysfunction, including narcolepsy syndrome, transient cognitive impairment and subacute rhombencephalitis/brainstem encephalitis; (3) delayed response, including focal brain necrosis and mild to moderate cognitive impairment, occurring months to years after radiation. Late delayed reactions are irreversible and destructive and, therefore, require significant attention [24]. Irradiation of the brain with different doses may induce neuroinflammation, apoptosis, reduced cell proliferation and differentiation and epigenetic and other histopathological changes, such as loss of endothelial cells and gliosis [25,26] (Figure 1). Radiation can damage DNA directly through ionization or indirectly through the generation of reactive oxygen species, thus inducing single-strand breaks (SSBs), alkaline oxidation, depurine or pyrimidine sites and, most importantly, double-strand breaks (DSBs). It may also lead to multiple non-DSB injuries, known as aggregation DNA damage [27,28,29]. Mitochondria consume 90% of the body’s oxygen and are the most abundant source of reactive oxygen species (ROS). Neuronal exposure to ionizing radiation can lead to oxidative events, and excessive reactive oxygen and endogenous reactive oxygen produced by radiation can lead to extensive and long-term mitochondrial DNA damage. Ionizing radiation induces mitochondrial damage in human neural precursor cells, which varies with the degree of DNA damage, the time of radiation exposure and the differentiation of cells [30,31,32]. After radiation exposure, mitochondrial DNA may be preferentially damaged or lost due to oxidative stress, followed by reduced respiratory chain activity and reduced mitochondrial function. A continuous increase in mitochondrial oxidative stress after exposure to ionizing radiation may lead to mutations [33,34]. The mechanisms of brain damage induced by radiation are complex and varied, mainly include oxidative stress, inflammation and DNA damage.

### 2.1. Oxidative Stress

Radiation exposure causes a wide range of neuronal damage, including increased oxidative stress, loss of neural stem cells and damage to neuronal structures [35]. It can also cause changes in calcium signaling cascades, significant activation of free radical processes, overproduction of reactive oxygen species in living cells, changes in neural and cognitive functions and destruction of the BBB [36]. Oxidative stress is a state of imbalance between the formation of reactive oxygen species and the antioxidant capacity of cells due to enhanced production of reactive oxygen species and/or dysfunction of the antioxidant system [37]. The production of ROS is mainly dependent on enzymatic and non-enzymatic reactions. The mitochondrial respiratory chain and various enzyme reactions are the endogenous sources of reactive oxygen species. Ionizing radiation is one of the exogenous activation sources of ROS. Non-enzyme free radicals are produced during mitochondrial respiration when cells are exposed to ionizing radiation [38]. At high concentrations and prolonged exposure, ROS destroy large molecules of cells, such as DNA, proteins and lipids, leading to cell necrosis and apoptosis [39]. The central nervous system (CNS) is inherently susceptible to oxidative stress and is highly active in oxidative metabolism, resulting in relatively high intracellular production of O_2_^−^ and other ROS. In some CNS regions and cells, especially oligodendrocytes, the iron content is very high, which can catalyze ROS production through the Fenton reaction (Figure 2). The Fenton reaction is a reaction between divalent iron ion and hydrogen peroxide which produces hydroxyl radical, strong oxidation ability [40]. The CNS is limited in its ability to perform aerobic glycolysis, so it is very vulnerable to hypoxia [41]. It contains relatively low levels of superoxide dismutase (SOD), catalase and glutathione peroxidase (GPX) [42] and low levels of antioxidants in oligodendrocytes, neurons and endothelial cells [43]. ROS play an important role in cell proliferation, cell movement, cell cycles and apoptosis [44]. The radiolysis of extracellular water during radiotherapy produces ROS, including superoxide anions (O_2_^−^), hydroxyl radicals (OH^−^) and hydrogen peroxide (H_2_O_2_), which are highly reactive entities harmful to normal tissue [45]. In addition, radiation can induce mitochondria to produce endogenous ROS [46] and change the permeability of mitochondrial membrane, thus further stimulating the production of ROS [47,48]. Excessive ROS can also destroy components of the electron transport chain in mitochondria and induce an imbalance in the intracellular redox system [49]. It also causes oxidative stress by reacting with biomolecules, such as lipids, proteins and DNA, leading to lipid peroxidation, protein misfolding and DNA strand breaking.

### 2.2. Inflammation

One of the earliest physiological responses of the body to ionizing radiation is the production of ROS [49], and while the physicochemical and free radical events are over within a microsecond of radiation exposure, the inflammatory response that ensues perpetuates the response by generating recurring waves of ROS, cytokines, chemokines and growth factors with associated inflammatory infiltrates [50]. Inflammation can cause sustained damage to various organs after exposure to radiation [51]. Radiation can change the number and function of immune system cells, increase macrophage and lymphocyte counts, induce secretion of inflammatory mediators, such as nuclear factor kappa-light-chain-enhancer of activated B cells (NF-κB) and small mother against decapentaplegic(SMAD)2/3, and cytokines, such as interleukin-1(IL-1), IL-2, IL-6, IL-13, IL-33, tumor necrosis factor α (TNF-α), transforming growth factor β (TGF-β), and interferon γ (IFN-γ) [52]. Exposure to high doses (more than 1 Gy) results in substantial DNA damage and cell death, leading to the release of cellular contents, such as danger alerts or damage-associated molecular patterns (DAMPs). The innate immune system senses pathogens and damaged cells through the DAMPs of pattern recognition receptors (PRRs), the most important type of PRRs being Toll-like receptors (TLRs). TLRs detects multiple DAMPs, and the most important DAMPs released after radiation exposure include high mobility group Box 1 (HMGB1), uric acid and heat shock proteins (HSPs) in the high mobility group [53,54]. TLR2, TLR4, TLR5 and TLR9 are the primary pattern recognition receptors in radiation reactions [55,56]; by recognizing DAMPs, they play a central role in the activation of inflammatory pathways [57,58]. Upregulation of NF-κB and other transcription factors leads to the secretion of pro-inflammatory cytokines [59]. NF-κB is a well-known radiation-responsive transcriptional and pro-inflammatory factor involved in inflammatory responses, cell cycle progression, apoptosis, and cell adhesion. Radiation-induced NF-κB activation has been reported in bone marrow, lymph nodes, spleen and intestine after total body irradiation [60]. Increased expression of pro-inflammatory cytokines begins immediately after exposure and can persist for months or even years. The response of the central nervous system to stressors and injuries (e.g., ionizing radiation) is mediated by the concomitant response of the innate immune effector cells of the brain, microglia. Exposure of brain tissues to high doses of ionizing radiation will lead to the expression and release of pro-inflammatory cytokines and reactive oxygen species in the brain [3], which, in turn, will lead to dysfunction or apoptosis of mature or newly differentiated cells, which integrate into the hippocampus network and show long-term functional defects [61]. Exposure to ionizing radiation activates microglia, which initiate an inflammatory response by releasing pro-inflammatory factors, including cytokines and ROS [62,63]. Activated microglia release ROS, reactive nitrogen and pro-inflammatory cytokines, such as TNF-α, IL-1 and IL-6, through mechanisms such as the NF-κB pathway or activate nicotinamide adenine dinucleotide phosphate (NADPH) to mediate neuroinflammation through the mitogen-activated protein kinase (MAPK) signaling pathway [64,65,66]. Neuroinflammation is, therefore, considered to be related to various functions and pathological states of the central nervous system.

### 2.3. DNA Damage

ROS produced by radiation attack and destroy multiple targets, most critically DNA. Radiation can damage DNA both by direct ionization of the DNA itself and by indirect processes in which it reacts with radioactive free radicals (such as OH^−^, H^−^ O_2_^−^ and H_2_O_2_) produced in physiological aqueous solutions surrounding the DNA. The result of these processes is a large number of different types of DNA damage, including base damage of varying degrees of complexity: SSBs, DSBs, and DNA cross-linking [67,68]. DSBs are the most lethal form of radiation damage to cells and are caused by phosphate backbone breaks and single-strand breaks in DNA. Single-strand breaks (SSBs), which appear not only as a direct result of radiation but also during nucleotide excision repair of DNA or during normal replication, can be converted into DSBs if they are not rejoined. If double-stranded DNA is not repaired completely, mutations, cancer and cell death will occur [69]. It is widely believed that radiation-induced genomic mutations may be related to DNA damage, which is converted into mutations during the processing of DNA repair mechanisms, or into inheritable mutations during DNA replication [70]. Ionizing radiation destroys many subcellular structures, extending from the plasma membrane to the nucleus, and radiation-induced cytotoxicity is closely related to DNA damage. The amount of radiation-induced DNA damage varies not only with the amount of ionization and their proximity to the double helix but also with the cell’s ability to scavenge free radicals and the efficiency of the DNA damage repair pathway [71]. DSBs are considered to be a complex form of DNA damage, which is the result of aggregation of multiple damage types [72]. There are many mechanisms of DNA repair, including base and nucleotide excision repair, mismatch repair, DNA damage bypass, non-homologous end joining (NHEJ), homologous recombination, single-strand annealing and cross-linked repair. However, NHEJ and homologous recombination are the main mechanisms of DSB repair [73]. DSBs are an initiator of genomic instability induced by radiation [74]. Genomic instability activates DNA damage responses, and individual DNA damage triggers different, but overlapping, pathways to regulate the cell cycle [75,76] and affects numerous distal functions, including DNA repair, transcription, cell cycle progression, apoptosis and senescence. DNA damage was caused by sensor protein identification, such as Mre11-Rad50-Nbs (MRN) complex and Ku70/80, which quickly activate the proximal signaling kinases A-T mutated (ATM), Rad3-related (ATR) and DNA-dependent protein kinase (DNA-PK) [77]. ATM is proven to regulate various cellular processes, including DNA repair, cell cycle progression, oxidative stress levels and mitochondrial homeostasis [78,79]. Countless downstream effector proteins are affected by ATM and cell cycle regulators, such as Caspase-2, Caspase-3 and p21. p21 acts as a downstream protein of p53 and plays a particularly critical role in regulating the cell cycle [80]. The yield of DSBs increases linearly with radiation dose. In addition, the more relaxed the DNA (as in transcriptional active DNA), the higher the production of DNA damage and the more complex it becomes [81].

## 3. Radio-Neuro-Protective Roles of Flavonoids

Flavonoids are chemically based upon a fifteen-carbon skeleton consisting of two benzene rings (A and B) linked via a heterocyclic pyrane ring (C) (as shown in Figure 3). According to the connection between the B-ring and C-ring, the structure of the B-ring and the hydroxylation and glycation patterns of the three rings, flavonoids can be divided into different subclasses [82], including flavanols, flavanones, flavonols, flavones, isoflavones and anthocyanins [83]. The chemical structure of flavonoid compounds depends on the spatial configuration of the hydroxyl group at the C2 position, the C2=C3 double bond, the 4-carbonyl group and the B-ring. The biological effects of flavonoids depend on their chemical structure. The position of hydroxyl groups and other features are important for their antioxidant and free-radical-scavenging effects [84]. Flavonoids reduce radiation-induced brain damage in different ways in many studies in cell and animal models. In the adult male rat model, pre-irradiation administration of quercetin with 50 mg/kg for 15 days can significantly improve malondialdehyde levels and total antioxidant status in plasma and tissues after acute cranial radiation with 20 Gy. Histopathological evaluation showed that quercetin administration significantly reduced radiation-induced neuronal degeneration and inflammatory infiltration, suggesting a neuroprotective role of quercetin after brain radiation exposure [22]. In primary cultured dorsal root ganglion (DRG) neurons, quercetin pretreatment (5–100 μM) for 24 h before gamma irradiation (2 Gy) resulted in increased viability, whereby viability increased in a quercetin-concentration-dependent manner. The maximum protective activity of quercetin against radiation-mediated toxicity was observed at concentrations of 25 and 50 μM. It reduced the expression of the endoplasmic reticulum stress marker gene in irradiated DRG neurons and downregulated the expression of TNF-α. Furthermore, it significantly increased the expression of the Tuj1 protein, suggesting neuronal revival, and decreased the apoptotic markers C/EBP-homologous protein (CHOP), jun-*N*-terminal kinases (JNK) and pJNK protein levels [85]. No toxicity was reported at the concentration used [22,85]. However, quercetin is a well-known flavonoid with low bioavailability. Recently, quercetin nanoparticles have been shown to have a better bioavailability [86]. In 6-week-old male albino Sprague–Dawley rats, pre-irradiation oral administration of Rutin with 200 mg/kg b.w/day for 21 days provided neuroprotection against γ-radiation-induced neurotoxicity via activation of the phosphatidylinositol-3-kinase (PI3K)/serine-threonin protein kinase (AKT)/ glycogen synthase kinase 3 β (GSK3β)/ nuclear factor erythroid-2 related factor-2 (NRF-2) pathway by altering the phosphorylation state through its ability to scavenge free radicals generation, modulating gene expression, and its anti-inflammatory effects when animals were γ-irradiated with 5 Gy [87]. In male Wister rats weighing about 120–150 g, pre-irradiation oral administration of 5, 7-dihydroxyflavone (DHF), a natural plant flavonoid, with 50 mg/kg b.w/day for 21 days can improve the content of malondialdehyde, β-amyloid, acetylcholinesterase, cysteine aspartic proteinase-3 and other parameters in the brains of rats γ-irradiated with 5 Gy [88]. In 7-week-old male C57BL/6 mice, pre-irradiation administration of baicalein (10 mg·kg·day-1 i.p.) for 7 days starting on postnatal day 42 showed that baicalein prevented radiation-induced necrotic death of neural progenitor cells and subsequent learning and memory retention deficits, suggesting that it may be a promising therapeutic candidate to protect radiation-induced impairment of neurogenesis and its neurocognitive consequences [23]. The pretreatment of irradiated rats with EGCG at doses of 2.5 and 5 mg/kg/d for 3 d before γ-irradiation with 4 Gy significantly ameliorated the increased plasma levels of amyloid β, TNF-α and IL-6 and the decrease in dopamine and serotonin. It also significantly ameliorated the irradiation-induced decrease in antioxidants, including glutathione level, and the activities of glutathione peroxidase and glutathione reductase in the hippocampus. EGCG treatment prior to radiation exposure protected against DNA damage and apoptosis in the hippocampus. Meanwhile, it reduced p53, Bax and caspases 3 and 9 but increased Bcl-2 expression. These results indicate that EGCG can attenuate the severity of radiation-induced damage and cell death in the hippocampus [89]. However, the inherent instability of EGCG limits its bioavailability and effectiveness. When formulated as dual-drug loaded nanoparticles (NPs) of EGCG/ascorbic acid (EGCG/AA NPs), EGCG displayed increased stability. Both EGCG and EGCG/AA NPs induced tight junction disruption and opened the BBB in vitro and ex vivo. Mechanistically, this study suggested that stabilization of EGCG in NP complexes and disruption of the BBB may result in higher therapeutic EGCG concentrations in the brain [90]. Oral administration of wogonin (5,7-dihydroxy-8-methoxy flavone) (30 mg/kg) for 15 days to adult male Wistar rats before or after acute whole-body γ-irradiation significantly reduced the TNF-α, IL-1β and IL-6 levels, whereas glutathione (GSH), SOD, catalase (CAT), GPX, NRF2, heme oxygenase-1 (HO-1) mRNA and protein expression were increased when compared with the irradiated group treated with distilled water. Subsequent neurodegeneration and gliosis were also reduced [91], suggesting that wogonin may also be a promising natural radioprotectant used as a complementary treatment to brain radiotherapy or accident radiation exposure (Table 1). It needs to be emphasized that some flavonoids have poor aqueous solubility, low bioavailability and extensive gastrointestinal and/or hepatic first-pass metabolism, leading to a quick elimination as well as low serum and tissue concentrations. The intranasal route may serve as a viable alternative to oral or parenteral administration. Flavonoids could be transported directly into the brain through the olfactory and trigeminal nerves without passing through the blood–brain barrier, and peripheral exposure is also reduced, which may minimize possible adverse effects [92].

## 4. Mechanisms of Flavonoids as Radio-Neuro-Protectants

Flavonoids have been well established to be antioxidant, scavenging free radicals, anti-inflammatory and maintaining genomic stability (Figure 4).

### 4.1. Antioxidant Activity of Flavonoids

Flavonoids have strong antioxidant activity due to their special chemical structure. The oxidation potential of flavonoid compounds is low, which is easy to be reduced by ROS, and they have a stronger free radical scavenging ability than high oxidation potential [96]. The difference in flavonoids scavenging ROS can be attributed to the different number and types of functional groups attached to the main nucleus. The C2=C3 double bond, adjacent conjugate ring C4-carbonyl group and catechol half by 4′5′-dihydroxy B-ring structure contribute to increased ROS scavenging. The antioxidant activity of flavonoids depends, in part, on their ability to delocalize electron distribution, resulting in a more stable phenoxy group. When flavonoid compounds react with free radicals, the phenoxy group produced is stabilized by the resonance action of aromatic nucleus [97]. Qurecetin contains almost all the functional groups needed for antioxidant activity and is therefore more effective than other flavonoids, such as catechins and hesperidins. Flavonoids can act as antioxidants in several different ways. First, they inhibit the production of free radicals. The production of free radicals in vivo includes enzymatic reaction and non-enzymatic reaction. Enzymatic reactions, such as xanthine oxidase (XO), lipoxygenase (LOX) and aldehyde oxidase (AO), are oxidases that can catalyze the generation of free radicals. Flavonoids reduce the production of free radicals by inhibiting the activity of these oxidases. Using 26 flavonoid compounds isolated from different plants, a study of the relationship between the structure of a flavonoid and its inhibitory ability of XO indicated that the double bond between hydroxyl on positions C5 and C7 and C2=C3 of flavonoid compounds significantly enhanced its inhibitory ability of XO [98]. Second, they directly eliminate free radicals. Flavonoids react with multiple hydroxyl groups in their molecules to form stable semi-quinone substances and terminate the chain reaction of free radicals. For example, baicalein and tea polyphenols can effectively eliminate ·O_2_^−^ [99,100]. Anthocyanins can also efficiently clear the free radicals and have an immunostimulating effect in preventing radiation-induced immunosuppression [93]. Several investigations have reported that silymarin alleviated irradiation-induced damage, including changes in nucleic acids and histone proteins, and inhibited radiation-induced free radical generation and lipid peroxidation [94,101]. Third, they activate the body’s antioxidant system. Endogenous antioxidant systems protect against radiation-induced oxidative stress by scavenging free radicals. For example, O_2_ can be converted to H_2_O_2_ by SOD, while catalase and peroxidase can convert H_2_O_2_ to water and O_2_ [102]. More specifically, the major determinants for radical-scavenging capability are (i) the presence of a catechol group in ring B, which has better electron-donating properties and is a radical target, and (ii) a 2,3-double bond conjugated with the 4-oxo group, which is responsible for electron delocalization. The presence of a 3-hydroxyl group in the heterocyclic ring also increases the radical-scavenging activity, while additional hydroxyl or methoxyl groups at positions 3, 5 and 7 of rings A and C seem to be less important. These structural features increase the stability of the aroxyl radical, i.e., the antioxidant capacity of the parent flavonoid. Thus, flavonols and flavones containing a catechol group in ring B are highly active, with flavonols more potent than the corresponding flavones because of the presence of the 3-hydroxyl group [103]. Since radiation can induce the reduction in enzyme-promoted antioxidant levels in cells, the improvement of antioxidant status in the flavonoid pretreatment process of cells can further reduce the attacks of free radicals on DNA, membrane lipids and other biological molecules, thus reducing the harmful effects of radiation on cells and tissues [84].

### 4.2. Flavonoids Reduce Inflammation in CNS

The brain is considered as “an immune-privileged organ”. Exposure to ionizing radiation will lead to premature neurovascular degeneration, increased death of cerebral microvascular endothelial cells, sparse cerebral microvessels and increased blood–brain barrier permeability [104]. Immune cells and markers of inflammation in the bloodstream cross the blood–brain barrier, and white blood cells infiltrate into CNS, triggering neuroinflammation [105]. Neuroinflammation is characterized by the activation of glial cells, including astrocytes and microglia, which represent primary immune cells in the brain [106]. Microglia, once activated, are involved in an inflammatory response that promotes the release of cytokines and chemokines, NF-κB, TNF-α, TGF-β, adhesion molecules and enzymes, such as 5-lipoxygenase (5-LOX), 12-LOX and cyclooxygenase-2 (COX-2) [107]. The molecular mechanisms of the anti-inflammatory activity of flavonoid compounds include inhibition of pro-inflammatory enzymes, such as cyclic oxygenase-2, lipopolysaccharide and inducible nitric oxide synthase, inhibition of NF-κB and activated protein-1 (AP-1) and activation of the second-stage antioxidant detoxification enzyme, MAPK, protein kinase C and Nrf2 [108,109,110]. Flavonoids also regulate protein kinases by inhibiting transcription factors, such as NF-κB [111]. This transcription factor regulates several cytokines, chemokines and cellular adhesion molecules involved in inflammation. Quercetin is a flavonoid found in a variety of fruits and vegetables, especially in the skin of onions. It reduces radiation-induced skin fibrosis. Quercetin and its derivatives have anti-inflammatory effects [112] and reduce oxidative stress [113]. Quercetin also influences NF-κB activation through extracellular signal-regulated kinase and p38 kinase inhibition, leading to radioprotection [114]. The inhibition of nitric oxide production by apigenin, genistein and luteolin is mediated by the downregulation of inducible nitric oxide synthase. Inflammatory cytokines produced and regulated at the transcriptional level can enhance or inhibit the inflammatory process [115]. Flavonoids inhibit the expression of inflammation-related enzymes/proteins, in part by inhibiting the activation of NF-κB, AP-1 and MAPK [108]. AP-1 transcription factors associated with inflammation and AP-1 signal transduction are important targets of flavonoid action [116,117]. Dietary flavonoids can also modulate inflammatory responses by activating pathways that induce antioxidant transcription and detoxification defense systems, such as glutathione peroxidase, heme oxidase, glutamyl cysteine synthase, superoxide dismutase and glutathione reductase, through antioxidant reaction elements [118,119,120]. Exposure to radiation can strongly affect the response of the immune system. Radiation can change the number and function of immune system cells. The increased number of macrophages and lymphocytes induces the secretion of various inflammatory mediators, and the increase in these inflammatory mediators is related to the release of prostaglandins and free radicals. Certain flavonoids can reduce the release of histamines or prostaglandins by mast cells or inhibit the production of pro-inflammatory cytokines or chemokines in mast cells, neutrophils and other immune cells [121,122]. Flavonoids inhibit dendritic cell maturation by inhibiting expression of mature markers, such as CD80/CD86 [123], and reduce the proliferation of CD4+ T cells. They also reduce the release of histamines, prostaglandins and cytokines by mast cells and bind to cytokine receptors to reduce signal transduction [124]. Flavonoids play an anti-inflammatory role mainly by inhibiting inflammation-related enzymes and immune pathways, reducing the inflammatory response of the CNS caused by radiation.

### 4.3. Flavonoids Reduce DNA Damage

DNA is a major target for radiation damage because it is the largest macromolecule in cells [80]. In vitro study has shown decreased DNA viscosity when exposed to X-rays due to a reduced molecular weight of DNA after DNA strand break [125]. Radiation can directly ionize DNA itself or damage it indirectly through processes in which DNA reacts with radioactive free radical product produced by physiological aqueous solutions around it, resulting in different DNA damages, such as DNA single- and double-strand breaks, modifications of deoxyribose rings and bases, intra- and inter-strand DNA–DNA cross-links and DNA–protein cross-links, and cause genetic mutations and disease [126,127,128]. Radiation produces reactive oxygen species that attack and destroy multiple targets, most critically DNA. If DNA damage is not properly repaired, it can lead to genomic instability, mutation and/or cell death. There are several mechanisms to reduce genomic instability caused by radiation, including scavenging free radicals, increasing the expression of endogenous enzymes and stabilizing the DNA double-helix structure. Flavonoids are not only antioxidant [103] and radioprotective but also anti-inflammatory [129] and anti-genotoxic [130]. Because flavonoid compounds have recognized antioxidant activity, the reduction in oxidative damage can further reduce DNA damage. The phenol ring of a flavonoid can specifically bind to bases or other groups on the DNA backbone and capture ROS. These polyphenols greatly reduce the effects of ionizing radiation at the molecular, cellular and/or tissue levels [84]. Genistein and quercetin have a variety of cellular effects, such as protecting normal cells from DNA damage [131]; genistein has immunomodulatory and radiation protection properties. By accelerating the recovery of neutrophils and platelets and the reconstruction of hematopoietic progenitor cells, genistein significantly increased the survival rate of γ-irradiated normal mice [132,133]. Flavonoids have a protective effect on genomic stability. Treatment of irradiated mice with flavonoids significantly reduced the level of primary DNA damage (SSBs). Naringin, epicatechin, and troxerutin all have this protective effect [95,134,135,136]. Flavonoids reduce cell death due to their antioxidant effects and protective effects on DNA. The interaction between flavonoids and DNA is one of the main mechanisms to protect DNA from tissue damage caused by radiation. Flavonoid intercalation into DNA has been established by spectroscopic and electrochemical methods. Evidence for intercalation comes from a major reduction in the intensity of UV–Vis bands’ characteristics of flavonoids upon DNA interaction [137]. The insertion of flavonoids into the DNA double-helix induces the stabilization of the DNA helix and condenses the DNA into a highly dense form that is less vulnerable to free-radical attacks. Positively charged flavones have stronger stabilization of double-stranded DNA than neutral flavones [138,139]. Flavonoids, which lack positively charged side groups, interact primarily with the phosphate backbone of DNA through hydroxyl groups [140]. The planar and aromatic moieties of flavonoids also contribute to DNA intercalation. Flavonoids’ rings A and C fuse together, providing a flat molecule that can be inserted between the stacked nucleic acid bases [141]. About 65 percent of DNA damage is caused by radiation, and the main role of flavonoids is to clear these free radicals. The improvement of antioxidant status during flavonoid pretreatment can further reduce the attacks of free radicals on DNA, membrane lipids and other biomolecules, thus reducing the harmful effects of radiation on cells and tissues.

## 5. Conclusions

High-dose ionizing radiation exposure to the brain has a negative impact on different brain cells, including neural stem cells, neurons, glial cells and endothelial cells, and promotes changes in the brain’s oxidation and inflammatory microenvironment. Overproduction of reactive oxygen species in the brain causes DNA damage, activates microglia cells to release inflammatory mediators, induces genomic instability and genetic mutations and increases the risk of cancer and hereditary diseases. Flavonoids can clear excess free radicals and stabilize the DNA double-helix structure due to their unique chemical structures. Flavonoids may also modulate and inhibit inflammation-related enzyme activity to produce anti-inflammatory effects and alleviate radiation-induced brain injury. These effects of flavonoid compounds depend entirely on their structure, especially the position of the hydroxyl group and the C2=C3 double bond. Because of the unique structure of flavonoids, these bioactive molecules may be promising radio-neuro-protective candidates for clinical use as a complementary treatment with brain radiotherapy or for radiation protection. While flavonoids have great radio-neuro-protective potential, their poor absorption, rapid metabolism and systemic elimination limit their bioavailability and may compromise their clinical use. Advances in nanotechnology have provided flavonoids which do not easily cross the BBB, a new route to reach effective concentrations in the central nervous system. The newly developed brain-targeted drug delivery pathways through the nasal cavity or the olfactory and trigeminal nerve may make the clinical use of flavonoids as radio-neuro-protectants promising. Future challenges lie in a deeper characterization of their therapeutic mechanisms and in the development of effective, safe and brain-targeted delivery systems for their intranasal administration.

## Figures and Tables

**Figure 1 molecules-25-05719-f001:**
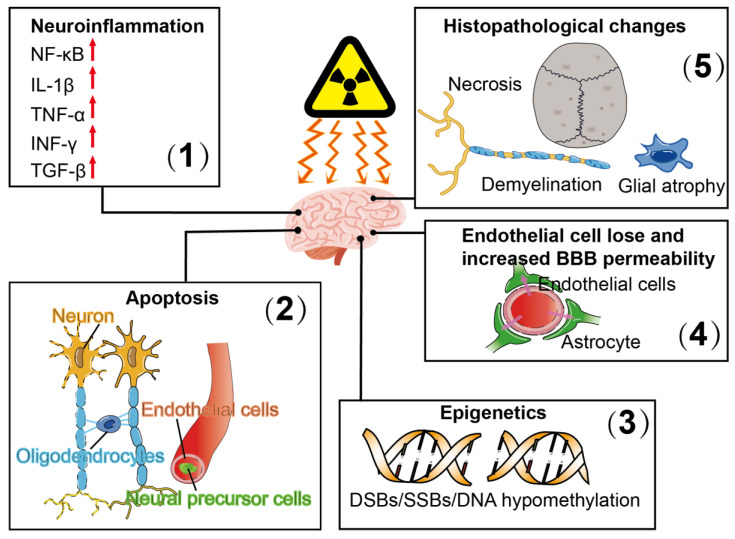
Pathophysiological responses of radiation-induced brain injury include (1) neuroinflammation, which is associated with increased expression of the transcription factor, nuclear factor kappa-light-chain-enhancer of activated B cells (NF-κB), as well as upregulated expression of interleukin-1 (IL-1)β, tumor necrosis factor α (TNF-α), interferon γ (INF-γ) and transforming growth factor β (TGF-β); (2) apoptosis of oligodendrocytes and neurons; (3) epigenetic alterations which are associated with DNA double-strand breaks (DSBs), DNA single-strand breaks (SSBs) and DNA hypomethylation; (4) endothelial cell loss and increased blood–brain barrier (BBB) permeability and (5) histopathological changes, including cell necrosis, glial atrophy and demyelination.

**Figure 2 molecules-25-05719-f002:**
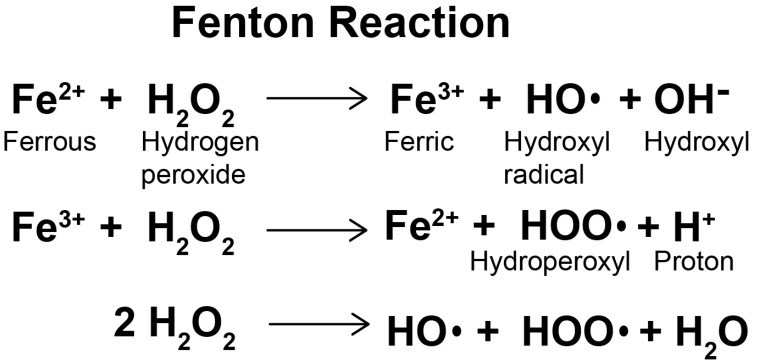
The Fenton reaction.

**Figure 3 molecules-25-05719-f003:**
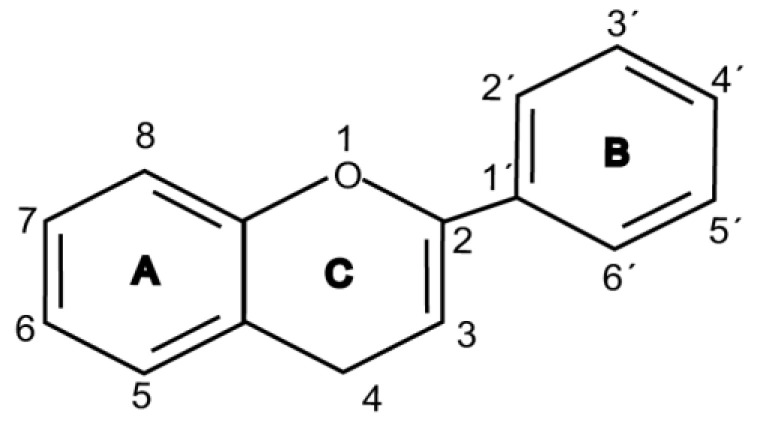
Basic structure of flavonoid.

**Figure 4 molecules-25-05719-f004:**
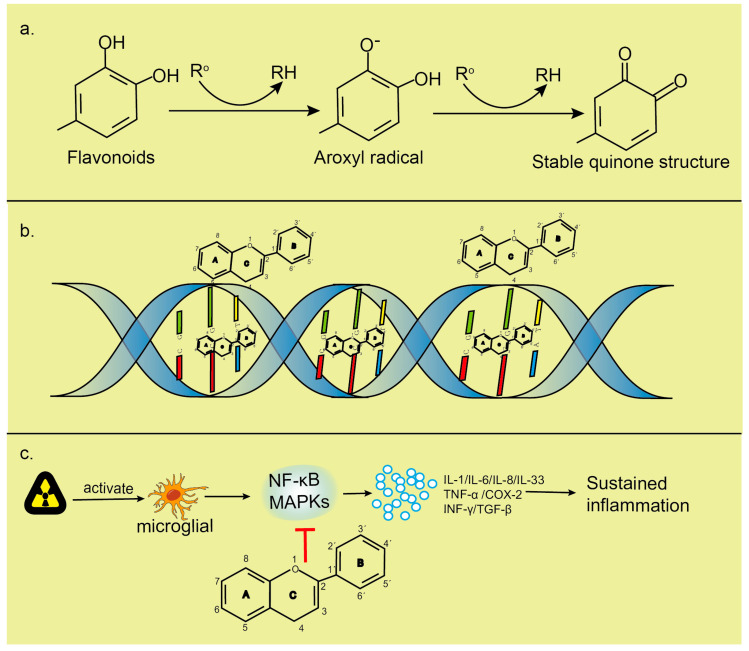
(**a**). Flavonoids are able to reduce highly oxidizing free radicals with redox potentials, such as superoxide, peroxyl, alkoxyl and hydroxyl radicals, by hydrogen atom donation. R^o^ represents superoxide anion, peroxyl, alkoxyl and hydroxyl radicals. The aroxyl radical may react with a second radical, acquiring a stable quinone structure. (**b**). Intercalation of flavonoids into DNA double helices induces stabilization of DNA helical structures and condensation of DNA to a highly compact form that is less susceptible to attacks by free radicals; flavonoids can interact with the phosphate moiety of the DNA backbone through hydrogen bonding. The repair of sugar radicals is attributed to hydrogen donation from flavonoids through this bonding. (**c**). Flavonoids inhibit the activation of NF-κB and MAPK, reduce the release of inflammatory factors and play an anti-inflammatory role.

**Table 1 molecules-25-05719-t001:** Classification, structure backbone and examples of the main classes of flavonoids.

Flavonoid Class	Structure Backbone	Example	Concentration	Radiation Type/Dose	Model	Mechanism	Ref.
Flavonol	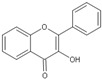	Quercetin	50 mg/kg/d	CT (20 Gy)	Rat	Antioxidant	[21]
			5–100 μM	γ ray (2 Gy)	Neuron	Downregulates TNF-α	[85]
Flavone	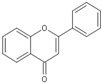	Baicalein	1–10 μM	γ ray (16 Gy)	neural progenitor cell	Antioxidant; neuroprotective	[22]
			10 mg/kg/d	γ ray (5 Gy)	Mouse		
Flavanol	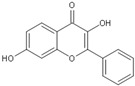	EGCG	2.5 and 5 mg/kg/d	γ ray (4 Gy)	Rat	Downregulates TNF-α, IL-6; protects hippocampus	[89]
Anthocyanin	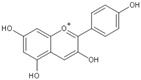	Cyanidin	200, 100 and 50 mg/kg/d	γ ray (6 Gy)	Mouse	Against immuno-suppression induced by the radiation	[93]
Flavanone	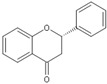	Silymarin	140 mg/kg/d	γ ray (0.2 and 0.6 Gy/d)	Rat	Repairs DNA damage	[94]
Isoflavone	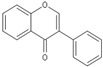	Genistein	200 mg/kg/d	γ ray (8.75 Gy)	Mouse	Protects the hematopoietic progenitor cell	[95]

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
