# Peer review of "Radioprotective Effect of Flavonoids on Ionizing Radiation-Induced Brain Damage"

_molecules, 2020, doi:10.3390/molecules25235719_

Round 1
Reviewer 1 Report
I do not recommend the review for publication in Molecules.
The review is supposed to cover the radioprotective effect of Flavonoids on radiation-induced brain damage. However, I don’t find that brain injuries are explained in particular. The authors have rather enumerated the radioprotective effect of flavonoids at the cellular level but, do this necessarily happen in the brain? Are flavonoids able to cross the BBB? Or are the authors assuming that the disruption created by radiation on the BBB help flavonoids to cross it?
In addition, it is not clear to what type of radiation the authors are referring to. In the introduction, which by the way does not really introduce the topic, they refer to the high levels of radiation that the population is nowadays exposed to (coming from different sources like nuclear power plants, mobile devices, industrial processes, UV radiation or hospital treatments). Then, do the authors want to explain the preventive effects of flavonoids to the high levels of radiation we are now exposed to? By the tittle and the abstract, one may expect to read about the effect that flavonoids have on brain injuries that are being treated (i.e. treatment of gliomas with radiotherapy or proton therapy).
The case studies of the use of flavonoids are also not clear. The authors have instead enumerated the properties of certain flavonoids but it is not clear if these properties have been demonstrated in vitro or in vivo, and more importantly if they actually have a protective effect to the exposure to radiation. What studies have been done? Have they been done in vitro, in animals or in patients? Likewise, how have the flavonoids been dosed? Actually, the authors keep referring to natural sources. Are there studies demonstrating the radioprotective effect of certain foodstuff? They may have antioxidant properties but do these help in brain injuries or prevent the development of brain injuries caused by radiation?
The structure of flavonoids is wrongly described in line 188-189. Ring C is not an oxygen-containing heterocycle!
Line 195: “The location and other characteristics of the hydroxyl group…” Location should be position, but what do the authors mean with other characteristics?
Line 334-336: Are the authors suggesting that intercalation of flavonoids into DNA protects it from the damage created by free radicals?
The lack of Figures and Schemes throughout the manuscript makes very difficult to follow the content. For example, the authors should be able to make a reaction scheme showing the antioxidant activity of flavonoids (section 4.1). How do they react with free radicals?
Punctuation is terrible.
Conclusions are incomprehensible.
Author Response
Journal: Molecules
Manuscript ID: 983857
Title: Radioprotective effect of Flavonoids on radiation- induced brain damage
Author(s): Qin Qi Wang, Cheng Hao Xie, Shijun Xi, Feng Qian, Xiao Chun Peng, Jiang Rong Huang and Feng Ru Tang
Dear Reviewer:
First, I would like to express my deep gratitude to you for the constructive comments and suggestions. We have completely revised the manuscript and addressed all the concerns raised by you.
The point-to-point responses to the comments by you have been attached for your perusal.
Best regards
Dr. Feng Ru TANG,
National University of Singapore
- “The review is supposed to cover the radioprotective effect of Flavonoids on radiation-induced brain damage. However, I don’t find that brain injuries are explained in particular. The authors have rather enumerated the radioprotective effect of flavonoids at the cellular level but, do this necessarily happen in the brain? Are flavonoids able to cross the BBB? Or are the authors assuming that the disruption created by radiation on the BBB help flavonoids to cross it?”.
Reply:
The explanation of the radiation-induced brain injury has been added the “Introduction” and Figure 1. We have also added references in the “Introduction” to address that flavonoids are able to cross the BBB to produce neuroprotection.
- “In addition, it is not clear to what type of radiation the authors are referring to. In the introduction, which by the way does not really introduce the topic, they refer to the high levels of radiation that the population is nowadays exposed to (coming from different sources like nuclear power plants, mobile devices, industrial processes, UV radiation or hospital treatments). Then, do the authors want to explain the preventive effects of flavonoids to the high levels of radiation we are now exposed to? By the tittle and the abstract, one may expect to read about the effect that flavonoids have on brain injuries that are being treated (i.e. treatment of gliomas with radiotherapy or proton therapy)”.
Reply: In this paper, we focused on ionizing radiation. To make it clear, we have revised the title of the manuscript as “ Radioprotective effect of Flavonoids on ionizing radiation-induced brain damage” .
- “The case studies of the use of flavonoids are also not clear. The authors have instead enumerated the properties of certain flavonoids but it is not clear if these properties have been demonstrated in vitro or in vivo, and more importantly if they actually have a protective effect to the exposure to radiation. What studies have been done? Have they been done in vitro, in animals or in patients? Likewise, how have the flavonoids been dosed? Actually, the authors keep referring to natural sources. Are there studies demonstrating the radioprotective effect of certain food stuff? They may have antioxidant properties but do these help in brain injuries or prevent the development of brain injuries caused by radiation?
Reply: Your concerns has been addressed in “3. Radio-neuro-protective roles of Flavonoids”. In general, most of studies are in cell and animal models.
- “The structure of flavonoids is wrongly described in line 188-189. Ring C is not an oxygen-containing heterocycle!”
Reply: The structure of flavonoids is revised.
- For “Line 195: “The location and other characteristics of the hydroxyl group…” Location should be position, but what do the authors mean with other characteristics?
Reply: “The location and other characteristics of the hydroxyl group…” has been revised in “3. Radio-neuro-protective roles of Flavonoids”.
- For “Line 334-336: Are the authors suggesting that intercalation of flavonoids into DNA protects it from the damage created by free radicals? The lack of Figures and Schemes throughout the manuscript makes very difficult to follow the content. For example, the authors should be able to make a reaction scheme showing the antioxidant activity of flavonoids (section 4.1). How do they react with free radicals?
Reply: We have revised Figure 3 to include a diagram to show how flavonoids react with free radicals.
- For “Punctuation is terrible.”
Reply: The manuscript has been re-written according to your suggestion.
- For “Conclusions are incomprehensible.”
Reply: The “Conclusion” was revised according to your suggestion.
Reviewer 2 Report
This is a review article focused on the radiation-induced brain damage and protective effects of flavonoids. I think this is a suitable article for publication in Molecules, but extensive revision for English language and readability is required. It is hard to read the sentences and comprehend what authors are trying to communicate. Some sentences just did not make any sense. Some sample suggestions related to section 2, and 2.1 are below, but these are by no means the only edits, and I advise the authors to revise the entire manuscript.
Abstract: Abstract does not effectively capture the gist of the manuscript. I think the first 7 lines in the abstract should be condensed, and elaborate the last sentence, to briefly detail the mechanisms.
Line 40-45: this is a very long sentence and incorrect grammar.
Line 58, Mechanisms of radiation-induced brain injury: It would be helpful if a schematic depicting various mechanisms of radiation induced brain injury can be illustrated in a figure (similar to Fig. 3).
Line 80, "...neuropathologic...": perhaps "...neuronal..." might fit better? "Neuropathologic" didn't make sense, although I understand what authors want to say.
Line 88, "...enzyme reactions of various enzymes...": This is a bad construction. Perhaps "...various enzyme reactions..." might fir well.
Line 90, "...also...": Remove. Across the manuscript, "also" is used very generously and must carefully be considered.
Line 91, "In high concentrations...": This should be "At high concentrations..."
Line 93-95: Revise this sentence. Its bad language.
Lines 95-98: Cannot understand the sentence.
Author Response
Journal: Molecules
Manuscript ID: 983857
Title: Radioprotective effect of Flavonoids on radiation- induced brain damage
Author(s): Qin Qi Wang, Cheng Hao Xie, Shijun Xi, Feng Qian, Xiao Chun Peng, Jiang Rong Huang and Feng Ru Tang
Dear Reviewer:
First, I would like to express my deep gratitude to you for the constructive comments and suggestions. We have completely revised the manuscript and addressed all the concerns raised by you.
The point-to-point responses to the comments by you have been attached for your perusal.
Best regards
Dr. Feng Ru TANG,
National University of Singapore
Reviewer 3 Report
I was honored to review the manuscript entitled “Radioprotective effect of Flavonoids on radiation-induced brain damage” submitted to Nutrients. The study presents high quality and deals with important clinical issue, such type of study is needed. I have only few small remarks that authors should address properly.
I recommend accepting the manuscript after minor revision.
There are only some points to correct:
- please provide the list of abbreviations
- introduction and discussion section need improvement – please provide information on how your results will translate into clinical practice; ex.g DOI: 10.3390/ijms21197306 ; DOI: 10.3390/jcm9020469
- It would be also useful to illustrate some of the explained mechanisms as in general the manuscript is quite long and therefore a little bit difficult to follow.
- in discussion section please provide study strong points and study limitation section
- please correct typos
I recommend accepting the manuscript after minor revision.
Author Response
Journal: Molecules
Manuscript ID: 983857
Title: Radioprotective effect of Flavonoids on radiation- induced brain damage
Author(s): Qin Qi Wang, Cheng Hao Xie, Shijun Xi, Feng Qian, Xiao Chun Peng, Jiang Rong Huang and Feng Ru Tang
Dear Reviewer:
First, I would like to express my deep gratitude to you for the constructive comments and suggestions. We have completely revised the manuscript and addressed all the concerns raised by you.
The point-to-point responses to the comments by you have been attached for your perusal.
Best regards
Dr. Feng Ru TANG,
National University of Singapore
- “please provide the list of abbreviations”
Reply: The list of abbreviations has been added in the revised version.
- “introduction and discussion section need improvement,please provide information on how your results will translate into clinical practice; ex.g DOI: 10.3390/ijms21197306 ; DOI: 10.3390/jcm9020469”,
Reply: The “Introduction” and “Discussion” have been revised according to the reviewer’s suggestions.
- “It would be also useful to illustrate some of the explained mechanisms as in general the manuscript is quite long and therefore a little bit difficult to follow.
Reply: Figure 1 has been prepared according to the Reviewer’s suggestion.
- “in discussion section please provide study strong points and study limitation section””,
Reply: Relevant part has been added in the “Discussion”.
- For “please correct typos”,
Reply: The manuscript has been checked carefully, and all the typos were corrected in the revised version.
Reviewer 4 Report
The manuscript is not easy to follow. The authors need to contact a native English speaker or somebody with more proficiency writting articles to improve substantially the article. When you read it, you have the feeling that you have read similar sentences previously in the article, and some sentences have no sense.
In the Fig 1 there is a mistake (oxygen heterocycle label is in the wrong position; benzene ring label is also in a wrong position). In Fig 2, six compounds out of ten have very serious mistakes. I do not understand this, because even if the authors are not chemists it is easy to find the right structures (for example in wikipedia).
Author Response
Journal: Molecules
Manuscript ID: 983857
Title: Radioprotective effect of Flavonoids on radiation- induced brain damage
Author(s): Qin Qi Wang, Cheng Hao Xie, Shijun Xi, Feng Qian, Xiao Chun Peng, Jiang Rong Huang and Feng Ru Tang
Dear Reviewer:
First, I would like to express my deep gratitude to you for the constructive comments and suggestions. We have completely revised the manuscript and addressed all the concerns raised by you.
The point-to-point responses to the comments by you have been attached for your perusal.
Best regards
Dr. Feng Ru TANG,
National University of Singapore
- “The manuscript is not easy to follow. The authors need to contact a native English speaker or somebody with more proficiency writting articles to improve substantially the article. When you read it, you have the feeling that you have read similar sentences previously in the article, and some sentences have no sense.
Reply: The manuscript has almost been re-written. We hope the revised version is acceptable.
- For “In the Fig 1 there is a mistake (oxygen heterocycle label is in the wrong position; benzene ring label is also in a wrong position). In Fig 2, six compounds out of ten have very serious mistakes. I do not understand this, because even if the authors are not chemists it is easy to find the right structures (for example in wikipedia).”
Reply: All the mistakes have been corrected in the revised version. Please check Fig. 2 and Table 1 in the revised version.

Round 2
Reviewer 1 Report
The authors have made a clear effort to address the comments regarding their review. However, there are still some aspects that need to be revised in order to improve the quality of the manuscript.
Major aspects:
- Line 76-80: Briefly mention why radiotherapy is the treatment option of head and neck tumors, what type of ionizing radiation can be used (X-rays, proton and carbon therapy) and why in spite of the tremendously improved dose conformity of intensity-modulated radiotherapy (IMRT) and fast ion-beam radiation, there is still a need to use radioprotective drugs to protect organs-at-risk.
- Please revise sentence in line 76 as there is no connection.
- Revise sentence 79-80: “complementary treatment with natural products…” It is confusing if the effect of natural products is improving the cytotoxicity of the radiotherapy treatment against malignant cells rather than exhibiting a protective effect to the surrounding healthy tissue.
- Line 181: Mention the Fentom reaction and illustrate with a scheme.
- Sentence line 269-272 makes no sense. Please rewrite.
- Section 3. It is not clear why the authors start explaining the activity of Quercetin to continue with Rutin and DHF and then again go back to Quercetin. And please be specific for all the cited examples and clarify how the flavonoid is added in vivo when possible (intravenously, intranasal??), and at what concentration they show the radioprotective effect. Do flavonoids exhibit any toxicity at the same concentration? When are flavonoid dosed, prior the radiotherapy treatment, simultaneously of after it? If different combination schemes have been studied, clarify which one presented the best outcomes.
- Table 1. It would make a difference if Table 1 includes some relevant data like: the concentration at which the flavonoid example shows the radioprotective effect, in combination to what type of radiation and radiation dose, what model was used to test te activity (in vitro -if so include cell types-, in vivo…), mechanism of protection (antioxidant…) and the corresponding reference.
- The authors mention in the conclusion how nanotechnology may help flavonoids to cross the BBB and to achieve the effective concentration in the CNS. This clearly exposes the limitations of flavonoids uptake in the brain together with a plausible remediation. I find this very interesting and I consider it worth discussing it in Section 3 (if related works are already published).
Minor aspects/Suggestions:
- Line 65-66: “Sustained exposure to low doses of radiation…” Maybe some example can be added in parenthesis like (e. mobile devices, X-rays, CT scans…)
- Line 109: Change “…and flavonoid is one of suitable candidates” to “…and flavonoids are suitable candidates”
- Line 123: “..may induces” to may induce
- Line 143: Is “oxidative events” suppose to be here?
- Caption of Figure 1: I suggest adding the abbreviation of DSB, SSBs…
- Line 181: Change “the production of ROS” for ROS production through the Fentom reaction.
- Line 189-190: “During radiotherapy, water in the extracellular environment produces ROS…” to “The radiolysis of the extracellular water during radiotherapy produces ROS…”
- Line 200-205: Rephrase it emphasizing that inflammation occur after ROS production.
- Line 221: Change “…[54],” to [54].
- Line 242: Rephrase as: “ROS produced by radiation attack…”
- Line 244: Change “free radical products” for “free radicals”
- Line 247: Change “…complexity,” for “…complexity:”
- Line 248-249:Is this correct “…and caused by phosphate backbone and SSBs in DNA”? What do the authors mean here.
- Line 275: “…Caspase-3, p21” to “…Caspase-3 and p21”
- Line 509: Change “…less susceptible to attack” for “…less susceptible to be attacked”
- Line 524: Change “…and stabilizing” for “…and stabilize”
- Line 530: I don’t think the position is random as it actually contributes to determine the biological activity of the flavonoids.
Reviewer 2 Report
This manuscript went through a total revamp and its readable in this version. I believe this review article is much improved, and I recommend its acceptance for publication in Molecules.
Author Response
Dear Reviewer:
Thanks very much for your approval of our manuscript, which is a great encouragement to us.
Best regards
Dr. Feng Ru TANG,
National University of Singapore
Reviewer 4 Report
The authors have improved considerably their previous manuscript. The formula of flavonoids have been corrected.
Figure 3A, the formula in the middle has a serious mistake because the carbon on top has 5 bonds. A degree student does not pass his/her course in Chemistry with that mistake. The vertical double bonds should be single. For the radical instead of º character the authors should use · character.
Some minor changes should be taken into account.
Flavonoid word though the text should not be with capital letter, but with lower case or minuscule.
Line 191: OH· instead of OH-
line 287: configuration instead of conformation
line 299: Rutin provides instead of "Rutin (a bioflavonoid) lavage treatment provides"
line 377: catechol instead of catechiol
line 379: "more stable phenoxy" group instead of "more stable phenoxyethylene group"
line 389: phenoxy instead of phenoxyethylene
line 474: "The phenol ring" instead of "The polyphenol ring
line 488: "flavonoids intercalate" instead "Flavonoid intercalates"
line 497: "a flat" instead "A flat"
